



# Icing Wind Tunnel Measurements of Supercooled Large Droplets Using the 12 mm Total Water Content Cone of the Nevzorov Probe

Johannes Lucke[1,2], Tina Jurkat-Witschas[1], Romy Heller[1], Valerian Hahn[1,3], Matthew Hamman[4],
Wolfgang Breitfuss[5], Venkateshwar Reddy Bora[6], Manuel Moser[1,3], and Christiane Voigt[1,3]

[1]Deutsches Zentrum für Luft- und Raumfahrt (DLR), Institute of Atmospheric Physics, 82234 Wessling, Germany
[2]Faculty of Aerospace Engineering, Delft University of Technology, 2629 Delft, Netherlands
[3]Institute of Atmospheric Physics, University of Mainz, 55881 Mainz, Germany
[4]Collins Aerospace, Uniontown, OH 44685, USA
[5]Rail Tec Arsenal, 1210 Vienna, Austria
[6]Institute of Fluid Mechanics, Technical University of Braunschweig, 38108 Braunschweig, Germany

**Correspondence:** Johannes Lucke (johannes.lucke@dlr.de)

**Abstract.** Supercooled large droplet (SLD) icing can occur behind the protected surfaces of an aircraft and create severe aerodynamic disturbances, which represent a safety hazard for aviation. Liquid water content (LWC) measurements in icing conditions that contain SLD require instruments that are able to sample unimodal and bimodal droplet size distributions with droplet diameters from 2 to $2000\,\mu m$. No standardized detection method exists for this task. A candidate instrument, that

is currently used in icing wind tunnel (IWT) research, is the Nevzorov probe. In addition to the standard 8 mm total water content (TWC) collector cone, a novel instrument version also features a 12 mm diameter cone, which might be advantageous for collecting the large droplets characteristic of SLD conditions. In the scope of the two EU projects SENS4ICE and ICE GENESIS we performed measurement campaigns in SLD icing conditions in IWTs in Germany, Austria and the USA. We obtained a comprehensive data set of measurements from the Hotwire, the 8 mm and 12 mm cone sensors of the Nevzorov probe

and the tunnel reference instrumentation. In combination with measurements of the particle size distribution we experimentally derive the collision efficiency curve of the new 12 mm cone for median volume diameters (MVDs) between 12 and $58\,\mu m$ and wind tunnel speeds from 40 to $85\,\mathrm{ms^{-1}}$. Knowledge of this curve allows us to correct the LWC measurements of the 12 mm cone ($\mathrm{LWC_{12}}$) in particular for the inevitably high decrease in collision efficiency for small droplet diameters. In unimodal SLD conditions, with MVDs between 128 and $720\,\mu m$, $\mathrm{LWC_{12}}$ generally agrees within $\pm 20\%$ with the tunnel LWC reference values

from a WCM-2000 and an Isokinetic Probe. An increase in the difference between $\mathrm{LWC_{12}}$ and the WCM-2000 measurements at larger MVDs indicates better droplet collision properties of the 12 mm cone. Similarly, the favorable detector dimensions of the 12 mm cone explain a 7% enhanced detection efficiency compared to the 8 mm cone, however this difference falls within the instrumental uncertainties. Data collected in various bimodal SLD conditions with MVDs between 16 and $534\,\mu m$ and LWCs between 0.22 and $0.72\,\mathrm{gm^{-3}}$ also show an agreement within $\pm 20\%$ between $\mathrm{LWC_{12}}$ and the tunnel LWC, which makes

the Nevzorov sensor head with the 12 mm cone the preferred instrumentation for measurements of LWC in Appendix O icing conditions.



*Copyright statement.* TEXT

## 1 Introduction

The fatal accident of an ATR-72 aircraft near Roselawn, Indiana in 1994 (National Transportation Safety Board, 1996; Marwitz
et al., 1997) prompted the Federal Aviation Administration (FAA) and European Union Aviation Safety Agency (EASA) to
review the existing regulations for flight in icing conditions. It also initiated numerous research activities which aimed to study
the occurrence and the distributions of supercooled large droplets (SLD), which are defined as droplets with diameters larger
than 100 µm. SLD mostly occur as part of bimodal droplet size distributions, i.e. a significant amount of small droplets is
present alongside the SLD (Cober and Isaac, 2012). Cober et al. (2009) separated SLD conditions into four subsets based
on the maximum drop size and the median volume diameter (MVD) of the droplet size distribution (DSD). Icing conditions
which contained drops with diameters larger than 500 µm were classified as freezing rain (FZRA) and conditions without drops
larger than 500 µm were classified as freezing drizzle (FZDZ). Furthermore, they distinguished between the conditions with
an overall MVD smaller than 40 µm (representing a strong small droplet mode) and those with an overall MVD larger 40 µm
(representing a strong large droplet mode). They also found that the occurrence of SLD conditions is in most cases limited to
a temperature range from -25 °C to 0 °C and to a relatively low liquid water content (< 0.44 g/m³) (Cober and Isaac, 2012).
Based on this analysis they developed an engineering standard that aircraft need to comply with in order to operate in SLD
conditions (Cober et al., 2009). This standard was eventually added to part 25 of the Federal Aviation Regulations (14 CFR)
and to EASA's certification specifications for large aeroplanes (CS-25) as Appendix O (Office of the Federal Register, National
Archives and Records Administration, 2016; European Aviation Safety Agency (EASA), 2021), hence the SLD conditions
which fall within its specifications are also called Appendix O conditions. Prior to the addition of Appendix O aircraft were
only certified for flying in icing conditions that fall into Appendix C of 14 CFR part 25 (Appendix C conditions). The droplet
distributions of Appendix C conditions consist of droplets with a mean effective diameter smaller than 50 µm and do not
contain SLD. Established instruments for measuring the liquid water content in Appendix C conditions include the King
probe (King et al., 1978), the WCM-2000 Multi-Element water content system (Steen et al., 2016) and the Nevzorov probe
(Korolev et al., 1998b, 2007, 2013; Schwarzenboeck et al., 2009; Strapp et al., 2003). In icing wind tunnels (IWTs) rotating
cylinders of various diameters (Stallabrass, 1978; Orchard et al., 2019) and icing blades (Ide, 1990) are used. In the absence of
standardized measurement methods many of these techniques are also employed to measure Appendix O conditions. However,
since the DSDs of Appendix O conditions span a significantly wider range of droplet sizes than Appendix C conditions, the
uncertainties associated with the measurement principles are significantly larger and have not been discussed in detail in the
literature, yet. In this work, we assess the performance of a Nevzorov probe in IWT conditions that contain SLD. Specifically
for the purpose of measuring SLD, a second, larger total water content (TWC) collector cone with a diameter of 12 mm has
been added to the Nevzorov sensor head (see Fig. 1), which is assumed to be appropriate for the collection of large droplets.
This large cone is also advantageous because its larger sample area provides better sampling statistics. However, due to its
larger size, the 12 mm cone also has a lower collision efficiency for small droplets, which has not yet been characterized. In





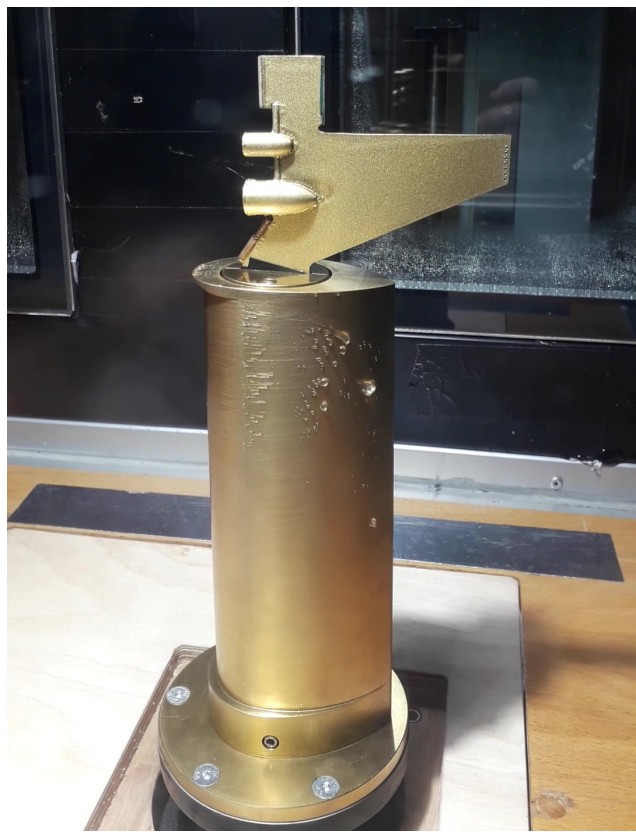

**Figure 1.** Nevzorov probe in the BIWT with the new sensor head, which features a hotwire (top), an 8 mm total water content collector cone (middle) and the new 12 mm total water content collector cone (bottom). The sensor head contains only one reference sensor which is positioned on the downwind side of the banneret that constitutes the top of the sensor head.

this work, we experimentally derive this collision efficiency and verify the new sensor's suitability to cover the large droplet size range of Appendix O conditions. The measurements on which we base our study were conducted in the scope of two EU-projects, which we introduce in the following section. Subsequently we describe the principle of operation of the Nevzorov probe, present the full set of measurements and derive a collision efficiency curve for the 12 mm cone. In the final sections we analyze the performance of the Nevzorov probe in unimodal and bimodal SLD conditions and investigate the errors that are

introduced when correcting for droplet collision efficiency with the MVD approximation.

## 2   The SENS4ICE and ICE GENESIS Research Projects

In the framework of Horizon 2020 the European Union funded two projects, SENS4ICE (SENSors and certifiable hybrid architectures for safer aviation in ICing Environment) and ICE GENESIS, with the goal to advance the capabilities of measuring, detecting and modelling SLD icing conditions and ice accretion. The SENS4ICE project aims to develop an airborne



**Table 1.** Specifications of the IWTs that were used for the measurements

| IWT | Test section size | Temperature range | Airspeed |
|---|---|---|---|
| Collins IWT | $152 \times 56 \times 112 \, \text{cm}^3$ | $-30°\text{C} - 0°\text{C}$ | $13 - 103 \, \text{m s}^{-1}$ |
| BIWT | $150 \times 50 \times 50 \, \text{cm}^3$ | $-20°\text{C} - 30°\text{C}$ | $10 - 40 \, \text{m s}^{-1}$ |
| Rail Tec Arsenal | $90 \times 2.5 \times 3.5 \, \text{m}^3$ | $-30°\text{C} - 5°\text{C}$ | $20 - 80 \, \text{m s}^{-1}$ |

hybrid ice detection system that is able to detect and differentiate between Appendix C and Appendix O conditions (Schwarz et al., 2019; Schwarz, 2021; Deiler, 2021). The system uses the measurements of direct icing sensors in combination with data that is obtained by monitoring the aircraft's flight characteristics (SENS4ICE, 2021). The ICE GENESIS project on the other hand focuses on developing advanced tools for the 3D simulation of SLD and snow icing conditions (ICE GENESIS, 2021). In both projects IWTs play a key role for the validation of the technology that is developed. The participating IWTs

consequently enhanced and adapted their spray system for Appendix O conditions. The production of Appendix O conditions is especially challenging because SLD sediment faster than smaller droplets and take longer to reach the freestream tunnel velocity and temperature (Orchard et al., 2018). Furthermore, the low LWC constraints of Appendix O complicate the generation of a continuous and homogeneous droplet spray (Ferschitz et al., 2017). In the framework of SENS4ICE, Appendix C and O conditions produced in three different IWTs were compared by measurements with the Nevzorov hot-wire probe and the Cloud

Combination Probe (CCP). Within ICE GENESIS, several campaigns were performed in the Rail Tec Arsenal (RTA) IWT with these and similar airborne instruments. We report on the measurements that were collected in Appendix C and Appendix O conditions as part of SENS4ICE and ICE GENESIS at the Goodrich IWT of Collins Aerospace in Ohio, the RTA IWT in Vienna, Austria and the Braunschweig IWT (BIWT) in Germany. Schematics of the three IWTs are shown in Fig. 2. The Collins IWT and the RTA IWT are well established facilities that have been involved in icing research for decades (Herman, 2006;

Collins Aerospace, 2021; Haller, 2005). Breitfuß et al. (2019) provide detailed information about the Appendix O conditions that are produced at RTA. The BIWT is a new facility whose design is described in (Bansmer et al., 2018). The tunnel was used for numerous research activities on ice crystal- and supercooled liquid water icing in recent years (Esposito et al., 2019; Knop et al., 2021). In 2019 and 2020 the tunnel spray system was upgraded to include the capability to create Appendix O conditions. All three wind tunnels have been calibrated per SAE ARP 5905 (AC-9C Aircraft Icing Technology Committee,

2015). For the characterization of the 12 mm Nevzorov TWC cone we evaluate measurements of LWC from these three tunnels in combination with the PSD measurements from the CCP and the tunnel reference instrumentation.

## 3 The Nevzorov probe's principle of operation

The Nevzorov probe is the primary instrument that we investigate in this work. Therefore we describe its principle of operation and the procedure to derive LWCs from its measurements. The Nevzorov probe belongs to the category of hot-wire instruments

(Korolev et al., 1998b). Such instruments contain heated sensing elements which are maintained at a constant temperature. Heat



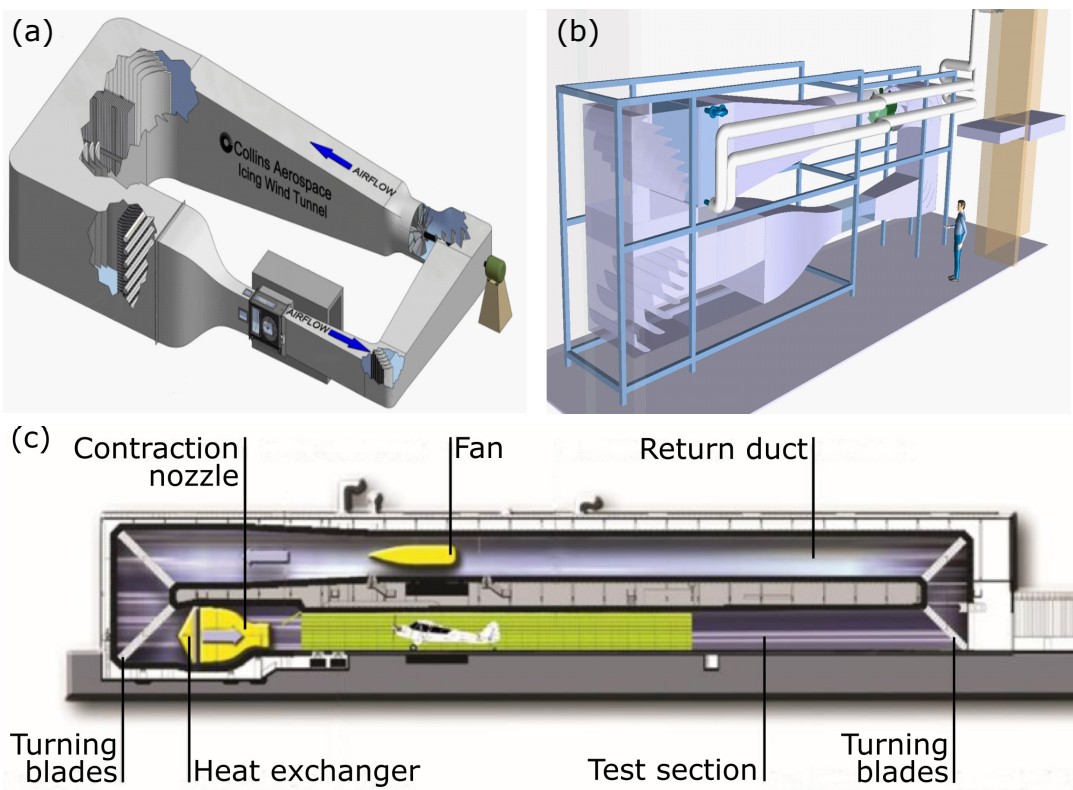

**Figure 2.** Schematics of the IWTs that were used for the measurements: a) Goodrich IWT of Collins Aerospace, b) Braunschweig Icing Wind Tunnel of the Technical University of Braunschweig, c) Climatic Wind Tunnel of Rail Tec Arsenal.

losses of these sensing elements are caused by convection and by impinging droplets which are heated and evaporated. From the power that is needed to maintain a constant temperature of the sensing elements the LWC and the TWC are estimated. In order to differentiate between convective heat losses and heat losses that are due to impinging water, the Nevzorov contains two types of sensors: Collector sensors are exposed to the airflow and the droplet spray. Their heat losses are due to evaporation and convection. The reference sensor on the other hand is protected from droplet impingement and its heat loss is solely due to convection. The Nevzorov probe outputs the voltages $V_c$ and currents $I_c$ of the collector sensors as well as the voltages $V_r$ and currents $I_r$ of the reference sensor. The power required by a collector sensor and a reference sensor is $P_c = V_c I_c$ and $P_r = V_r I_r$ respectively. Since the heat losses of a reference sensor are mainly due to convection its power consumption is assumed to be equal to (Korolev et al., 1998b):

$$P_r = \alpha_r S_r (T_r - T_a) \tag{1}$$

Here, $T_r$ and $T_a$ are the temperatures of the reference sensor and the ambient air, $S_r$ is the sample area of the reference sensor. The factor $\alpha_r$ is the heat transfer coefficient for the sensor, which in the literature is specified as $\alpha_r = K g_r N u_r$, where $K$ is the thermal conductivity of air, $g_r$ the factor which takes into account the surface geometry of the sensor and $Nu_r$ the





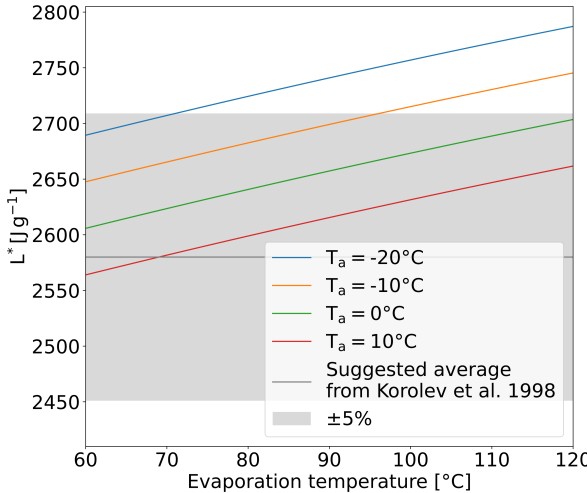

**Figure 3.** The energy needed to heat and evaporate water ($L^*$) plotted versus the evaporation temperature $T_\mathrm{e}$.

Nusselt number (Korolev et al., 1998b). In purely liquid clouds, the collector sensors need to heat the droplets from the droplet
temperature $T_\mathrm{d}$ to the evaporation temperature $T_\mathrm{e}$. The latent heat required for the evaporation at temperature $T_\mathrm{e}$ is $L(T_\mathrm{e})$. $T_\mathrm{d}$
can be assumed to be equal to $T_\mathrm{a}$, due to the dominance of the latent heat term even differences of 10°C between $T_\mathrm{d}$ and $T_\mathrm{a}$
would result into an error of less than 2% in the specific energy necessary for heating and evaporation ($L^*$):

$$L^* = c_\mathrm{w}(T_\mathrm{e} - T_\mathrm{a}) + L(T_\mathrm{e}) \tag{2}$$

Here, $c_\mathrm{w} = 4.1813\,\mathrm{J\,g^{-1}K^{-1}}$, which is the specific heat capacity of water. $L(T_e)$ can be approximated by the following formula
(Science Engineering Associates, 2016):

$$L(T_\mathrm{e}) = 2486.9696 - 2.025056 \cdot T_\mathrm{e} - 29.288 \cdot 10^{-4} \cdot T_\mathrm{e}^2 \; [\mathrm{J\,g^{-1}}] \tag{3}$$

The IWTs were unpressurized, hence $T_\mathrm{e}$ is equal to 100 °C. Korolev et al. (1998b) states a value of $2580\,\mathrm{J\,g^{-1}}$ as a good average
for the value of $L^*$, however Fig. 3 shows that this value is an underestimate for IWT conditions. The total power consumption
of the collector sensors is calculated by adding the convective term to the power required for heating and evaporating the
impinging water:

$$P_\mathrm{c} = \alpha_\mathrm{c} S_\mathrm{c}(T_\mathrm{c} - T_\mathrm{a}) + \varepsilon W L^* S_\mathrm{c} U \tag{4}$$

Here, $W$ denotes the water content of the air, $S$ is the sensor sample area, $U$ is the airspeed and $\varepsilon$ is the collection efficiency
of the sensor. A relation between the dry air losses of the reference sensor and the dry air losses of the collector sensor can be
obtained from measurements in dry air:

$$\frac{P_\mathrm{c,dry}}{P_\mathrm{r}} = k \tag{5}$$




The ratio $k$ only depends on parameters such as airspeed, altitude and temperature (Korolev et al., 1998b) and is thus constant for individual test points in IWTs. Rearranging and inserting Eq. (5) into Eq. (4) and solving for $W$ yields:

$$W = \frac{P_c - kP_r}{\varepsilon L^* SU} \tag{6}$$

The collection efficiencies that are required for solving Eq. (6) are partly available from the literature: The shape of the
hotwire sensor is approximately cylindrical and its collision efficiency can be calculated analytically as described in Finstad et al. (1988a) or Langmuir and Blodgett (1946). It is worth noting that these efficiencies only take into account the collision efficiency of droplets with the sensor. The overall collection efficiency of the hotwire sensor decreases once droplets reach sizes of 30-40 µm due to droplet splashing, as has been shown in Schwarzenboeck et al. (2009). Collision efficiencies of the 8 mm cone have been published by Strapp et al. (2003) based on a 2-D fluid simulation for velocities of 67 and 100 m/s. However,
these collision efficiencies likely contain significant errors for small droplet sizes. Splashing is assumed to be irrelevant for the 8 mm cone in Appendix C conditions (Strapp et al., 2003). For the 12 mm cone, which is a new addition to the Nevzorov sensor head, no collision efficiencies have been published up to now. In this paper, we experimentally derive the collision efficiency of the 12 mm cone from the measurements that we obtained during the wind tunnel campaigns.

## 4 IWT conditions and instrumentation

This chapter defines the IWT conditions that have been tested and the instruments, measurement principles and uncertainties. A remark on terminology: We differentiate here between small droplet spray (SDS), freezing drizzle (FZDZ) and freezing rain (FZRA). SDS includes the nominal Appendix C conditions (Jeck, 2002) as well as conditions where the LWC and MVD are outside the Appendix C envelopes, but no supercooled large droplet mode (D > 100 µm) is present. FZDZ and FZRA conditions include unimodal and bimodal SLD conditions, of which some fall within the LWC specifications of Appendix O
(Cober and Isaac, 2012) while others exceed the maximum LWC significantly. The distinction between FZDZ and FZRA is made according to the maximum of the LWC distribution in the large droplet mode, if the maximum is positioned at a diameter smaller than 500 µm we identify the condition as FZDZ, otherwise we identify it as FZRA. This definition is slightly different to that used in Cober and Isaac (2012), where FZRA is defined by the presence of droplets larger than 500 µm. The distribution of the droplet spray produced in the wind tunnels is relatively broad, so that sprays with a droplet mode centered around 200 µm
still contain a small, but not insignificant (>1% of total LWC) amount of droplets larger than 500 µm. We decided that such conditions are nonetheless better described by the characteristics of FZDZ as defined in Cober and Isaac (2012) and hence list them as such.

### 4.1 Instrumentation

Complementary to the Nevzorov LWC measurements, LWCs of all the test points used in this work have been measured
by the tunnel operators. These measurements are designated as the tunnel LWC and serve as a comparison to the Nevzorov measurements. For the tunnel LWC measurements the IWT operators employed a wide range of instruments, which we refer





to as the tunnel reference instrumentation. The tunnel reference instrumentation depends on the type of the produced droplet spray. Collins and RTA use icing blades to measure SDS conditions. The LWC for the SDS conditions of the BIWT were obtained with high accuracy flow meters, but the tunnel has previously also been calibrated with rotating cylinders and an

Isokinetic Probe (IKP) (Knop et al., 2021). In FZDZ conditions the BIWT again relies on flow meters, while Collins uses a WCM-2000. RTA computes its LWC in FZDZ conditions from the measurements of multiple instruments, among them icing blades, the WCM-2000 (King-Steen et al., 2021b; Steen et al., 2016), the Nevzorov probe (Korolev et al., 1998b) and the Cloud, Aerosol and Precipitation Spectrometer (CAPS) (Baumgardner et al., 2001, 2017). The LWC of FZRA conditions was determined solely from IKP measurements (Davison et al., 2012; Strapp et al., 2016; Ratvasky et al., 2021). An overview of

all the instrumentation used in the tunnels is shown in Table 2.

Beside the LWC measurements, DSDs obtained with airborne instrumentation were provided by DLR, Embraer and the wind tunnel owners. The DSDs constitute an important input parameter for the collision efficiency calculation of the Nevzorov probe. At the BIWT we measured the DSDs with the DLR HALO-CCP, which was flown during various flight campaigns (Voigt et al., 2017; Jurkat-Witschas et al., 2019; Voigt et al., 2022; Papke Chica et al., 2022) and has been described in Braga

et al. (2017a, b). The CCP used at Collins was provided by Embraer. For the measurements at RTA we use DSDs derived from data of an FCDP (Glienke and Mei, 2020; Kirschler et al., 2022), a 2D-S (Lawson et al., 2006), a PIP (Baumgardner et al., 2017), a CAPS that was provided by DLR and a Malvern Spraytec probe provided by the tunnel operator (Ferschitz et al., 2017).

**Table 2.** Tunnel reference instrumentation used by the IWT operators

| IWT | LWC reference instrumentation | | | | Droplet size reference instrumentation | | |
|---|---|---|---|---|---|---|---|
| | Small droplet spray | FZDZ | | FZRA | Small droplet spray | FZDZ | FZRA |
| Collins | Icing Blade | WCM-2000 | | | CCP | CCP | |
| RTA | Icing Blade | Icing Blade, WCM-2000, Nevzorov, CAPS | IKP | | Malvern | Malvern, FCDP, 2D-S, CAPS | Malvern, FCDP, 2D-S, PIP |
| BIWT | Flow meters | Flow meters | | | CCP | CCP | |

## 4.2 Measurements with the CCP

We now give an overview of the CCP and the data evaluation for the DSDs. The CCP consists of two instruments, the Cloud Droplet Probe (CDP) which measures droplet size based on the intensity of the forward scattered light and the Cloud Imaging Probe (CIP) which records the shadow images of droplets on its array of photo diodes. The CDP detects droplets in the size range from 2-50 μm and outputs data in bins with 1-2 μm bin width. We applied a size binning for liquid droplets based on a laboratory calibration to the lower end of the CDP size range in order to consider ambiguities caused by the Mie resonances

(Lance et al., 2010; Rosenberg et al., 2012).

The CIP measures particles in the size range from 15-950 μm with a size resolution of 15 μm. We processed its data with the SODA software (Bansemer, 2013). The software incorporates a shattering (Field et al., 2006) and a depth of field correction (Korolev et al., 1998a). For the combination of the measurements of CDP and CIP we defined a threshold within the overlap



region of the instruments at which we transitioned from using the CDP data to using the CIP data. The threshold value was
chosen in a way that ensured that the CDP provided sufficient sampling statistics. Depending on the number concentration of
droplets in the transition region it therefore varied between 39 μm and 47 μm. After combining the data of the two instruments
we followed the procedure in Cober and Isaac (2012) and performed a logarithmic interpolation between the bin centers to
obtain a size distribution with 1 μm bins.

### 4.3 Measurement uncertainties

All instruments are subject to measurement uncertainties, which we discuss now. For hotwire LWC measurement techniques
Baumgardner et al. (2017) state a propagated uncertainty of 10%-30% due to errors related to the removal of convective heat
losses and the uncertain response to large droplets and ice crystals. Convective heat losses depend on temperature, airspeed
and pressure. These parameters are held constant in an IWT and we observed that the errors in the dry air calibration of
the Nevzorov are generally well below 5% of the measured LWC. Only liquid water conditions were investigated hence
uncertainties due to the response to ice crystals are irrelevant. The accuracy of the probe itself is ±10% according to the
manufacturer (SkyPhysTech Inc., 2020). We therefore estimate an overall uncertainty within ±15%, as also stated by Korolev
et al. (1998b). For the WCM-2000 King-Steen et al. (2021a) found biases of 5-15% between two sensor heads, which were
caused by a misaligned calibration and an increased amount of solder on one of the sensing elements. For both instruments
these accuracy values apply for the size range of typical Appendix C conditions, whereas uncertainties in SLD conditions are
larger.

Uncertainties of accretion based methods such as the rotating cylinder and the icing blade are generally assumed to be low
in low LWC Appendix C conditions, Stallabrass (1978) states an absolute LWC accuracy within ±10% for both methods in
conditions with MVDs between 14 and 34 μm. Accretion based methods however have their limitations when high LWC or
large droplets are involved and uncertainties depend on the size of the element that is used (Steen et al., 2016; Orchard et al.,
200 2019).

For optical particle measurements we distinguish between the sizing and the counting accuracy. For instruments based
on light scattering, such as the CDP, the propagated uncertainty is 10%-50% for particle sizing, while the uncertainty in
concentration is 10%-30% (Baumgardner et al., 2017). For imaging probes, uncertainties generally may extend from 10%-
100% for both size and concentration (Baumgardner et al., 2017). For the CIP we performed an analysis of the uncertainty
in the measured number concentration based on a laboratory calibration and information from the literature. According to the
analysis, the uncertainty in the measured number concentration is smaller than 15% for droplets larger than 80 μm, but could
increase to 60% for droplets smaller than 80 μm.

Discrepancies between LWC values measured by different instruments may also arise from the mounting positions of the in-
struments. For instance, at the Collins IWT the Nevzorov probe was mounted 45 cm downstream of the WCM-2000 calibration
position. The measurements were performed individually with each instrument. There is also an inherent separation between
the sample volumes of some probes, on the Nevzorov sensor head the Hotwire and the 12 mm TWC cone are positioned ap-
proximately 2 cm above and below the 8 mm cone. On the CCP the separation between the sample volumes of CIP and CDP is





13.5 cm. Such differences in position are especially relevant when the spray homogeneity is poor, as might be the case for large droplet spray (Ferschitz et al., 2017; Orchard et al., 2018). For that reason, traverse measurements with the Nevzorov probe

were performed in the BIWT to determine the height at which the large droplet spray was concentrated.

### 4.4 IWT conditions

Table 3 provides an overview of all the test points from the three IWTs used for this study. At Collins, we measured a total of twenty-one SDS conditions at air speeds of 40, 67 and 85 $\mathrm{m\,s^{-1}}$. Eight different SDS conditions were measured in the BIWT at the maximum tunnel airspeed of 40 $\mathrm{m\,s^{-1}}$. At RTA, four SDS conditions were measured at a tunnel speed of 60 $\mathrm{m\,s^{-1}}$. The

FZDZ conditions vary significantly between the tunnels. Collins produced unimodal SLD conditions with MVDs between 128 and 221 μm at an airspeed of 76 $\mathrm{m\,s^{-1}}$. At RTA and the BIWT we measured mostly bimodal freezing drizzle distributions with varying fractions of LWC in the small and large droplet modes. Only test point U19 at RTA is unimodal. Currently, of the three IWTs, only RTA is able to produce freezing rain conditions. We obtained measurements in unimodal as well as bimodal freezing rain conditions at air speeds of 50 and 60 $\mathrm{m\,s^{-1}}$.

## 5 Derivation of collision efficiencies

The problem of droplet collision efficiency on various geometries has been thoroughly investigated in the literature (Langmuir and Blodgett, 1946; McComber and Touzot, 1981; Lozowski et al., 1983; Makkonen, 1984; Finstad et al., 1988a). A droplet trajectory can be described as a function of two parameters, the droplet inertia parameter $K$, which relates the droplet inertia to the drag forces that act on the droplet and the free stream droplet Reynolds number $Re$ (Heinrich et al., 1991). The two

parameters are specified in Eq. (7) and (8) respectively.

$$K = \frac{1}{9}\frac{D^2 V_\infty \rho_\mathrm{w}}{c\mu_\mathrm{a}} \tag{7}$$

$$Re = \frac{\rho_\mathrm{a} V_\infty D}{\mu_\mathrm{a}} \tag{8}$$

In the equations $D$ denotes the droplet diameter, $V_\infty$ the free stream velocity, $\rho_\mathrm{a}$ and $\rho_\mathrm{w}$ are the densities of air and water, $c$

is the characteristic length of the geometry for which the impingement is calculated and $\mu_\mathrm{a}$ is the dynamic viscosity of air. If the Reynolds number is held constant, droplet collision efficiencies increase with an increase in the droplet inertia parameter $K$ (Heinrich et al., 1991). Therefore, larger droplets, a larger airspeed and a smaller sensor geometry result in higher collision efficiencies. Consequently, we expect lower collision efficiencies for the 12 mm cone than for the Hotwire and the 8 mm cone.

One possibility to derive the collision efficiencies of the Nevzorov 12 mm sensor experimentally is to compare its mea-

surements in the IWT with a reference LWC value, measured with well-established sensors such as those listed in ARP5905 (AC-9C Aircraft Icing Technology Committee, 2015). Collision efficiencies curves can then be estimated with a fit through the data points. Such reference LWC values exist for the Appendix C conditions of the three IWTs reported here, however they



**Table 3.** Overview of the test points measured in the SENS4ICE and ICE GENESIS IWTs. The LWC values stem from the internal tunnel calibration. MVD values were derived from CCP measurements at the BIWT and Collins and from CAPS and Malvern measurements at RTA.

| **Collins IWT** | | | | | **Rail Tec Arsenal** | | | | | **BIWT** | | | | |
|---|---|---|---|---|---|---|---|---|---|---|---|---|---|---|
| Test point | TAS [m s$^{-1}$] | SAT [°C] | LWC [g m$^{-3}$] | MVD [$\mu$m] | Test point | TAS [m s$^{-1}$] | SAT [°C] | LWC [g m$^{-3}$] | MVD [$\mu$m] | Test point | TAS [m s$^{-1}$] | SAT [°C] | LWC [g m$^{-3}$] | MVD [$\mu$m] |
| Small droplet spray | | | | | Small droplet spray | | | | | Small droplet spray | | | | |
| C1 | 40 | -20 | 0.30 | 12 | LWC29* | 60 | 5 | 0.43 | 15 | 406 | 40 | -10 | 0.27 | 22 |
| C10 | 40 | -20 | 1.50 | 18 | LWC28* | 60 | 5 | 0.43 | 20 | 416 | 40 | -10 | 0.64 | 29 |
| C2 | 40 | -10 | 0.42 | 15 | LWC27* | 60 | 5 | 0.43 | 40 | 405 | 40 | -10 | 0.18 | 34 |
| C12 | 40 | -10 | 0.42 | 15 | LWC26* | 60 | 5 | 0.44 | 50 | 409 | 40 | -5 | 0.61 | 21 |
| C3 | 40 | 0 | 0.54 | 18 | | | | | | 410 | 40 | -5 | 0.55 | 26 |
| C11 | 40 | 0 | 2.50 | 16 | | | | | | 419 | 40 | -5 | 0.80 | 30 |
| C5 | 67 | -20 | 0.25 | 14 | | | | | | 418 | 40 | 0 | 0.82 | 26 |
| C14 | 67 | -20 | 0.80 | 27 | | | | | | 417 | 40 | 0 | 0.81 | 32 |
| C6 | 67 | -10 | 0.42 | 15 | | | | | | | | | | |
| C15 | 67 | -10 | 1.40 | 19 | | | | | | | | | | |
| C19 | 67 | -10 | 1.10 | 42 | | | | | | | | | | |
| C29 | 67 | -10 | 1.30 | 46 | | | | | | | | | | |
| C30 | 67 | -10 | 1.50 | 53 | | | | | | | | | | |
| C4 | 67 | 0 | 0.80 | 14 | | | | | | | | | | |
| C13 | 67 | 0 | 2.00 | 17 | | | | | | | | | | |
| C8 | 85 | -20 | 0.30 | 13 | | | | | | | | | | |
| C17 | 85 | -20 | 1.30 | 20 | | | | | | | | | | |
| C9 | 85 | -10 | 0.34 | 19 | | | | | | | | | | |
| C18 | 85 | -10 | 0.80 | 28 | | | | | | | | | | |
| C24 | 85 | -10 | 0.90 | 41 | | | | | | | | | | |
| C25 | 85 | -10 | 1.20 | 58 | | | | | | | | | | |
| Freezing Drizzle | | | | | Freezing Drizzle | | | | | Freezing Drizzle | | | | |
| O2 | 76 | -18 | 0.79 | 158 | U13†* | 40 | 5 | 0.22 | 24 | 522† | 40 | -5 | 0.72 | 16 |
| O3 | 76 | -18 | 1.08 | 221 | U15†* | 40 | 5 | 0.64 | 102 | 521† | 40 | -5 | 0.47 | 18 |
| O4 | 76 | -18 | 1.45 | 172 | U19* | 40 | 5 | 0.5 | 126 | 524† | 40 | -5 | 0.44 | 24 |
| O5 | 76 | -18 | 1.48 | 188 | U18†* | 60 | 5 | 0.43 | 102 | 525† | 40 | -5 | 0.38 | 34 |
| O6 | 76 | -18 | 1.66 | 152 | | | | | | 537†* | 40 | -5 | 0.36 | 61 |
| O7 | 76 | -18 | 1.65 | 128 | | | | | | | | | | |
| O8 | 76 | -18 | 1.51 | 153 | | | | | | | | | | |
| Freezing Rain | | | | | Freezing Rain | | | | | Freezing Rain | | | | |
| | | | | | TP10 | 50 | -5 | 0.30 | 720 | | | | | |
| | | | | | TP11 | 60 | -5 | 0.25 | 720 | | | | | |
| | | | | | TP7†* | 60 | 3 | 0.33 | 534 | | | | | |
| | | | | | TP8† | 60 | -5 | 0.33 | 534 | | | | | |

† Bimodal distribution

* For testing purposes the tunnel temperature was raised above the melting point.





**Table 4.** Number of small droplet spray measurements per airspeed group.

| IWT | Group 1 ($40\,\mathrm{m\,s^{-1}}$) | Group 2 ($60$ and $67\,\mathrm{m\,s^{-1}}$) | Group 3 ($85\,\mathrm{m\,s^{-1}}$) |
|---|---|---|---|
| Collins | 6 | 9 | 6 |
| RTA | 0 | 4 | 0 |
| BIWT | 8 | 0 | 0 |
| Total | 14 | 13 | 6 |

were measured with different instruments. Alternatively, the measurement of the Nevzorov Hotwire and the 8 mm cone can be used as a reference. Since the collection efficiencies of these two sensors are well characterized, they can be corrected and be

a measure for the true tunnel LWC. The advantage of using these two sensors as a reference is that they measured the exact same condition that the 12 mm cone was subjected to, hence the comparison is not affected by random fluctuations of LWC in the IWTs.

Since the collision efficiency curve changes with airspeed, we define three groups of measurements in Appendix C conditions, which can be seen in Fig. 5 and Table 4. Group 1 contains measurements at $40\,\mathrm{m\,s^{-1}}$, Group 2 contains measurements

at 60 and $67\,\mathrm{m\,s^{-1}}$ and Group 3 contains measurements at $85\,\mathrm{m\,s^{-1}}$ from Collins IWT. The measurements of Group 1 contain measurements from all three tunnels. Group 2 contains measurements from RTA and Collins which also differ in airspeed by $7\,\mathrm{m\,s^{-1}}$. We group these measurements together because we assume that the gain in accuracy of the collision efficiency curve that we obtain from using more measurements outweighs the inaccuracy that we induce by not differentiating between the air speeds.

We compute the LWC that was present in the tunnel from the Hotwire and the 8 mm cone measurements of the Nevzorov. As mentioned before, large droplets tend to splash on the Hotwire, whereas for the 8 mm cone the collision efficiency of small droplets is low, which makes the LWC estimate prone to large uncertainties. We use the appropriate sensor for each measurement; if the MVD is smaller or equal to $20\,\mu\mathrm{m}$ we utilize the collision-efficiency corrected Hotwire measurements, while for an MVD larger than $20\,\mu\mathrm{m}$ we use the collision-efficiency corrected 8 mm cone measurements ($\mathrm{LWC_8}$). From now

on we call this combination of LWC values from the Hotwire and the 8 mm cone the Nevzorov reference LWC and denote it as $\mathrm{LWC_{Nevz}}$. In an ideal experimental setup the Nevzorov probe would be exposed to monodisperse droplet distributions, the measurements of the 12 mm cone would be compared to $\mathrm{LWC_{Nevz}}$ and a collision efficiency curve could be derived. Realistic conditions differ to that setup because dispersed droplet distributions are produced. In our experiments, these droplet distributions are derived from the CCP or from the tunnel reference instrumentation. We assume the collision efficiency curve

of the 12 mm cone can be described by a function $\varepsilon_{12} = f(d)$, where $d$ is the droplet diameter. For an ideal measurement, the raw LWC measured by the 12 mm cone ($W_{12}$) is equal to the LWC in the tunnel, which we approximate with $\mathrm{LWC_{Nevz}}$, multiplied by the overall collision efficiency of the 12 mm cone (see Eq. (9)).





**Table 5.** $D_0$ values computed from the curve fit for the different airspeed groups. The uncertainties represent the $2\sigma$ intervals that are associated with the curve fit.

| Group | G1 | G2 | G3 |
|---|---|---|---|
| $D_0$ | $18.3 \pm 2.3$ | $18.7 \pm 1.1$ | $17.6 \pm 2.6$ |

$$W_{12} = \sum_{d_{\min}}^{d_{\max}} f(d_i) \cdot v(d_i) \cdot \mathrm{LWC_{Nevz}} \tag{9}$$

Here, $v(d_i)$ is the fraction of total LWC in size bin $i$, calculated from the available size distributions. The question arises what kind of analytical function $f(d)$ should be. Korolev et al. (1998b) suggested Eq. (10) for $f(d)$, where $D_0$ is the free parameter, which can be adjusted depending on the sensor that is modelled. We also experimented with other functional forms but found that Korolev's curve produced the best results.

$$f(d) = \frac{d^2}{\left(D_0^2 + d^2\right)} \tag{10}$$

In a next step we formulate a system of equations for each airspeed group, where each equation represents one measurement and is of the form of Eq. (9). We find the optimal solution for $D_0$ for each air speeds group with least squares estimation, which minimizes the sum of squared residuals $S$ (see Eq. (11)) with respect to $D_0$.

$$S = \sum_{T_1}^{T_n} \left( \frac{W_{12,\,T_j}}{\mathrm{LWC_{Nevz,\,T_j}}} - \sum_{d_{\min}}^{d_{\max}} f(d_i, D_0) \cdot v_{T_j}(d_i) \right)^2 \tag{11}$$

In the equation above, $T_j$ denote the individual test points. The results of the least squares estimation is shown in Table 5. Figure 4 shows the computed collision efficiency curves. The three curves for the three different airspeed groups lie very close together, so that they are hardly distinguishable. The collision efficiency of a $10\,\mu\mathrm{m}$ diameter droplet is only 0.2, but it then rises steeply to 0.5 for $20\,\mu\mathrm{m}$ droplets. Beyond $20\,\mu\mathrm{m}$ its slope decreases continuously and the collision efficiency attains 0.7 for $30\,\mu\mathrm{m}$ droplets and 0.9 for $60\,\mu\mathrm{m}$ droplets. We note that the collision efficiency curve for group 1 ($40\,\mathrm{m\,s^{-1}}$) is slightly higher than that of group 2 (60 to $67\,\mathrm{m\,s^{-1}}$). This is unexpected, because a higher airspeed leads to higher momentum and therefore results in a higher collision efficiency, in line with equations (7) and (8). However, within the stated error margins of the $2\sigma$ intervals, also the scenario $D_{0,\mathrm{G1}} < D_{0,\mathrm{G2}}$ is possible. Figure 5 shows the corrected LWC measurements from the 12 mm cone ($\mathrm{LWC_{12}}$) and $\mathrm{LWC_{Nevz}}$ for the SDS test points. Each row contains a different airspeed group. The left panels depict the ratio of $\mathrm{LWC_{Nevz}}$ to the tunnel LWC, i.e. they compare how well the reference measurements from the Nevzorov probe and the tunnel agree. The shaded areas denote 10% and 20% deviation from the tunnel LWC measurements. The comparisons show a good agreement between $\mathrm{LWC_{Nevz}}$ and the tunnel LWC, where, across all airspeed groups, 58% and 94% of the Nevzorov measurements fall within $\pm 10\%$ and $\pm 20\%$ of the tunnel LWC respectively. The scatter of the data points can therefore be explained through the combined uncertainties of the Nevzorov probe and the wind tunnel.





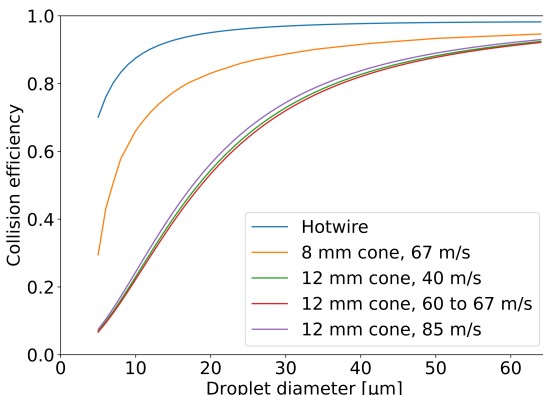

**Figure 4.** Collision efficiency curves of the 12 mm cone for the three airspeed groups, the collision efficiencies of the Hotwire and the 8 mm cone from the literature are shown for comparison.

The right panels show the ratio of $LWC_{12}$ to the tunnel LWC. For airspeed group 2 (Fig. 5d) $LWC_{12}$ exhibits a similarly good agreement to the tunnel LWC as $LWC_{Nevz}$ (Fig. 5c). For airspeed groups 1 and 3 (Fig. 5b and 5f), the discrepancies between $LWC_{12}$ and the tunnel LWC are a bit larger than between $LWC_{Nevz}$ and the tunnel LWC (Fig. 5a and 5e). Across all

airspeed groups, 42% and 79% of the $LWC_{12}$ values fall within $\pm 10\%$ and $\pm 20\%$ of the tunnel LWC respectively. The outliers at low MVDs are mostly data points with high LWCs. There has been an ongoing discussion concerning the Nevzorov's ability to evaporate all of the impinging water. For an earlier, shallower version of the Nevzorov's cone Emery et al. (2004) observed, that a pool of water formed inside the cone and was occasionally swept out, which led to an underestimate of the LWC. The effect occurred during ice shaver conditions run at an airspeed of $67\,\mathrm{m\,s^{-1}}$ and a TWC larger than $2.1\,\mathrm{g\,m^{-3}}$. For this work,

a thorough analysis of the data found no evidence of pooling. Pooling and subsequent underestimates of LWC should be a function of LWC flux. While $LWC_{12}$ is lower than the tunnel LWC for some of the high LWC flux test points, it is equal or higher for many others (see Fig. 5 and Table 3). The discrepancy between $LWC_{12}$ and the tunnel LWC for the low MVD and high LWC points can in part be explained through droplet coincidence effects in the CDP. The number concentrations for these test points exceeded $2000\,\mathrm{cm^{-3}}$ and droplet coincidence (Lance et al., 2010; Lance, 2012) was present. Droplet coincidence

results in a shift towards larger particles in the size distribution, which in turn decreases the applied collision efficiency. The magnitude of the effect and its exact influence on $LWC_{12}$ could not be determined, because the inter-arrival time data, which we use to correct for coincidence, was not available for the measurements at Collins.

## 6 Nevzorov probe measurements in unimodal SLD conditions

The Nevzorov probe was also tested in unimodal large droplet conditions, see Table 3. These test points provide valuable infor-

mation on the response of the Nevzorov sensors to large drops. Figure 6 shows the results of the measurements in comparison





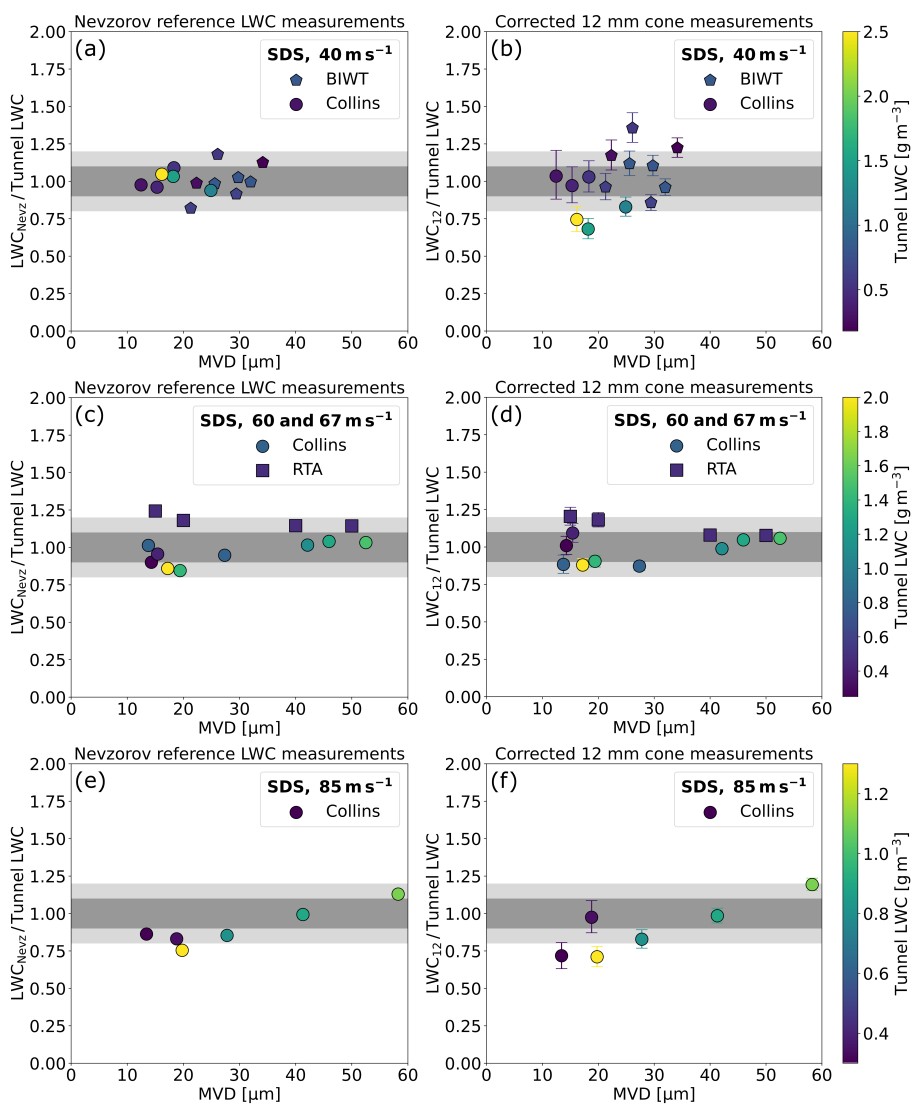

**Figure 5.** Comparison of the Nevzorov reference LWC and the corrected 12 mm cone LWC to the tunnel LWC for the three different airspeed groups. The error bars represent the uncertainty in the collision efficiency curve, that is described by the standard deviation of $D_0$ (see Table 5). They do not incorporate other uncertainty sources, such as biases in the Nevzorov's response or IWT variability. For high MVDs, the error bars are in some cases too small to extend outside the marker.




to the tunnel LWC, determined with the WCM-2000 instrument for the FZDZ cases (except for the lowest MVD FZDZ test point, that was measured at RTA with multiple instruments) and with an IKP for the FZRA test points. No collision efficiency corrections were applied to any of these measurements because the droplet diameters were deemed to be sufficiently large for collision efficiency effects to be irrelevant (hence $LWC_{12} = W_{12}$). The overall agreement between the Nevzorov and the tunnel

LWC is good, all $LWC_8$ measurements and all but one $LWC_{12}$ measurement fall within $\pm 20\%$ of the tunnel LWC. $LWC_8$ and $LWC_{12}$ generally follow a similar trend in comparison to the tunnel LWC, but the $LWC_{12}$ measurement is on average 6.5% higher than the $LWC_8$ measurement. For the FZDZ test points, where the tunnel LWC was determined with the WCM-2000, $LWC_8$ and $LWC_{12}$ increasingly exceed the tunnel LWC for increasing MVD values. This does not apply for the FZRA test points, for which the tunnel LWC was determined with the IKP.

The results suggest that the Nevzorov TWC sensors are better suited than the WCM-2000 for the collection of droplets with diameters of approximately $200\,\mu m$ or more. A possible explanation is the greater depth and width of the Nevzorov sensors, which allows them to retain most of the large droplets. Splashing and bouncing effects, similar to those described by Korolev et al. (2013) for an earlier, shallower version of the Nevzorov TWC cone might occur on the $2.1\,mm$ wide WCM-2000 TWC sensor. In line with these observations, a comparison of the WCM-2000 and the IKP shows that the LWC measurements of the

IKP exceeded those of the WCM-2000 (Lang et al., 2021), in FZRA conditions even by as much as 65%.

We remark, that there can be other factors which cause or contribute to the discrepancies between Nevzorov and WCM-2000, such as the different mounting positions of the two instruments or an uneven distribution of the large droplet spray. Also, the high LWC large droplet spray at Collins led to oscillations of the sensor head, which may have affected the result of the measurements.

The fact that $LWC_{12}$ is on average higher than $LWC_8$ suggests that the 12 mm cone is preferable to the 8 mm cone for the collection of large droplets, as its perimeter to area ratio is smaller than that of the 8 mm cone, the probability of droplet re-entrainment into the airflow after impacting inside the cone decreases. However, we note that the difference between the two cones is still within the uncertainty range of the instrument.

## 7 Application of collision efficiencies

We now apply the newly computed droplet collision efficiencies to bimodal distributions measured in the BIWT and the RTA wind tunnel. As Collins only produces unimodal DSD there is no data available from this IWT. An overview of cumulative liquid water content (CWC) from the bimodal DSDs measured with the CCP in the BIWT can be seen in Fig. 7. Often, collision efficiencies of DSDs are approximated by using the MVD as a representative diameter for the entire distribution. This has been shown to work well for small cylindrical sensors and unimodal droplet distributions (Stallabrass, 1978; Finstad et al., 1988b).

Recently, Sokolov and Virk (2019) found that Langmuir A-J distributions with similar MVDs had very different collision efficiencies on a 30 mm cylinder at an airspeed of $4\,m\,s^{-1}$. Furthermore, larger errors can be introduced when using the MVD approximation for bimodal distributions (FAA, 2014). Van Zante et al. (2021) also caution that bimodal distributions cannot be fully captured and represented by the MVD. We investigate the magnitude of the errors introduced by using the MVD





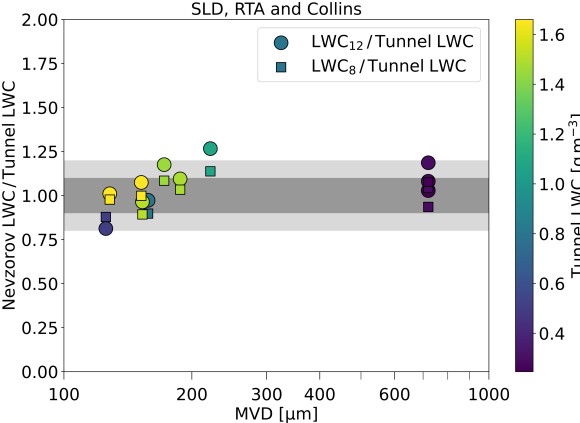

**Figure 6.** Measurements of the 12 mm cone and the 8 mm cone in comparison to the tunnel LWC in unimodal SLD conditions. The tunnel LWC is based on WCM-2000 measurements for all the FZDZ test points and on the IKP for the FZRA test points.

approximation for droplet collision efficiency for a number of bimodal distributions measured at in the BIWT, see Fig. (7).

The figure also shows the relative error in the LWC when the MVD approximation for droplet collision efficiency is used. It highlights the importance of using the entire size distribution for the computation of the collision efficiency, especially for sensors such as the 12 mm cone where a large collision efficiency correction is applied. In one bimodal distribution the error from the usage of the MVD approximation for droplet collision efficiency exceeded 30%. Note that the relative error is not a function of the MVD, but rather depends on how well the MVD represents the DSD.

In Fig. 8 and Table 6 we present a comparison of $LWC_{12}$ to the tunnel LWC for the bimodal FZDZ and FZRA conditions that we measured in the BIWT and at RTA. The $LWC_8$ is plotted for comparison. The results show that $LWC_{12}$ and $LWC_8$ agree within $\pm 20\%$ with the tunnel LWC for all but one test point. We also observe that the measurements of the two Nevzorov cones, $LWC_{12}$ and $LWC_8$, coincide closely with each other once the MVD exceeds $24\,\mu\mathrm{m}$. At lower MVDs, $LWC_{12}$ and $LWC_8$ diverge into opposite directions from the tunnel LWC.

The results prove that reliable measurements of LWC in bimodal SLD conditions can be achieved with the 12 mm TWC cone of the Nevzorov probe. The collision efficiency correction appears to be very accurate once the MVD exceeds $24\,\mu\mathrm{m}$. The divergence of $LWC_{12}$ and $LWC_8$ from the tunnel LWC at lower MVD can be seen as an indication that minor errors still exist in the collision efficiency curve of the 12 mm cone and possibly also in that of the 8 mm cone, as acknowledged by Strapp et al. (2003). The analytical form for the collision efficiency curve of the 12 mm cone is simple, therefore it is probable that

the curve cannot accurately represent the collision efficiency for all diameters. Furthermore, the collision efficiency at small diameters is low, so that even a small offset in the curve introduces large errors in the result. For the test point at an MVD of $61\,\mu\mathrm{m}$, both $LWC_{12}$ and $LWC_8$ exceed the tunnel LWC by approximately the same value and the offset is consistent for both measurements that were made in this condition. This indicates that the discrepancy is not due to a problem with the Nevzorov probe or the collision efficiency correction, but more likely a larger uncertainty in the tunnel calibration exists for this point.





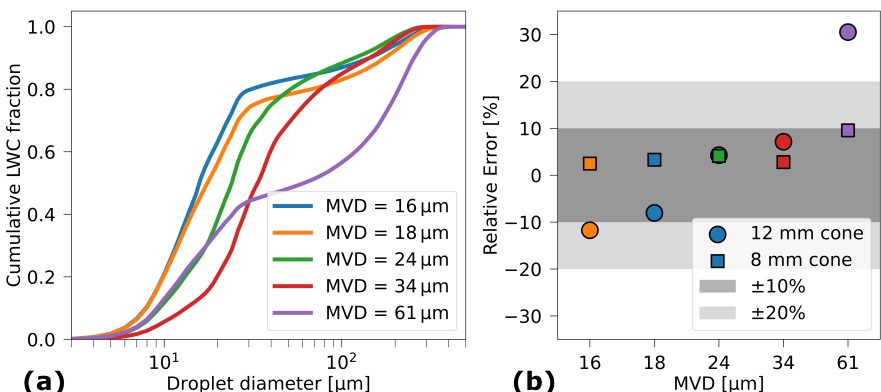

**Figure 7. (a)**: Cumulative liquid water content of the bimodal DSDs measured in the BIWT. The distributions with the MVDs of 16, 18, and 61 μm have a small droplet mode centered around 15 μm and a large droplet mode at approximately 230 μm. They differ mainly in the ratio of LWC contained in small droplets to LWC contained in large droplets. The distributions with MVDs of 24 and 34 μm have their small droplet mode centered at 20 and 30 μm respectively and their large droplet mode at 165 μm. **(b)**: Relative error of the LWC estimate that is introduced when approximating the collision efficiency from the MVD as compared to the collision efficiency estimated using the droplet size distributions shown on the left side.

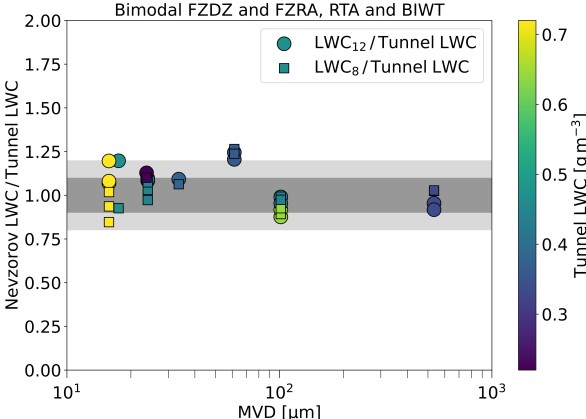

**Figure 8.** Same as Fig. 6 but for bimodal distributions and with collision efficiencies applied. For the BIWT FZDZ test points, the tunnel LWC is based on flow meter measurements. The tunnel LWC of the FZDZ test points from RTA was determined from a combination of icing blade and WCM-2000. The tunnel LWC of the FZRA test points from RTA stems from a combination of icing blade and IKP measurements. Table 6 lists the IWT where the individual test points originated.

Finally we would like to note that errors in the size distribution, that we use as an input for the computation of the collision efficiency, propagate into the errors of the LWC.





**Table 6.** Comparison of the $\mathrm{LWC_{12}}$ to the tunnel LWC and $\mathrm{LWC_8}$ The values of test points that were measured multiple times were averaged.

| Test point | Tunnel | MVD [μm] | $\mathrm{LWC_8}$/Tunnel LWC | $\mathrm{LWC_{12}}$/Tunnel LWC | $\varepsilon_{12mm}$ |
|---|---|---|---|---|---|
| 522 | BIWT | 16 | 0.94 | 1.07 | 0.49 |
| 521 | BIWT | 18 | 0.93 | 1.16 | 0.53 |
| 524 | BIWT | 24 | 1.02 | 1.06 | 0.61 |
| U13 | RTA | 24 | 1.10 | 1.10 | 0.66 |
| 525 | BIWT | 34 | 1.06 | 1.07 | 0.72 |
| 537 | BIWT | 61 | 1.25 | 1.21 | 0.70 |
| U15 | RTA | 102 | 0.91 | 0.89 | 0.85 |
| U18 | RTA | 102 | 0.98 | 0.95 | 0.84 |
| TP7 | RTA | 534 | 1.02 | 0.95 | 0.90 |
| TP8 | RTA | 534 | 1.02 | 0.91 | 0.90 |

## 8 Conclusions

This work investigates the performance of a new, 12 mm diameter TWC cone of the Nevzorov probe using data collected in three different IWTs. We compared the LWC measured with the 12 mm cone to the measurements of the Hotwire and the

8 mm cone of the Nevzorov probe as well as to the tunnel LWC. We found that a large correction needs to be applied to compensate for the low droplet collision efficiency of the cone. We experimentally derived this collision efficiency for three different air speeds using test points with MVDs between 12 and 58 μm. For the shape of the collision efficiency curve we prescribed the functional form suggested in Korolev et al. (1998b). In order to obtain the highest accuracy, we used the droplet size distributions of each individual test point for the derivation. We verified the capability of the 12 mm cone to collect SLD

through a comparison with the tunnel reference instrumentation, which included a WCM-2000 and an IKP. The results indicate that the 12 mm cone has better droplet collection properties than the WCM-2000 when the droplet size exceeds 200 μm. Even in FZRA conditions, the 12 mm cone does not suffer from any significant losses of LWC, instead our comparison showed a good agreement to the values of the IKP. The 12 mm cone also appears to be better suited for the collection of SLD than the 8 mm cone, because it measured slightly but consistently higher LWC values. The difference between the two cones is however

still within their mutual uncertainty range.

We subsequently applied the new collision efficiency correction to measurements collected with the 12 mm cone in bimodal distributions and compared the resulting LWCs to those of the 8 mm cone and the tunnel LWC. The comparison showed an agreement within ±20% with the tunnel LWC for all but one test point, highlighting the ability of the 12 mm cone to provide accurate measurements across the entire size range of Appendix O conditions. We observed that some inaccuracies remain

in the computed curves at small droplet diameters and caution should therefore be exercised when using the 12 mm cone in conditions that contain strong small droplet modes. For such conditions the collision efficiency curve for the 12 mm cone may



be applied but the corrected LWC readings should be compared to those of the Hotwire and the 8 mm cone to assess their plausibility.

We also investigated the magnitude of the errors that can be introduced when one approximates the collision efficiency via the MVD instead of using the entire size distribution. The error depends on the collision efficiency correction that is applied and on the size distribution. For the collision efficiency curve of the 12 mm cone the error exceeded 30% in one case, which stresses the importance of using the entire size distribution in the collision efficiency calculation.

In summary, our results and findings demonstrate that the 12 mm cone of the Nevzorov probe is suitable for the measurement of SLD icing environments. Future IWT- and flight campaigns with the Nevzorov will be able to use the 12 mm cone as a reliable source for the LWC with excellent properties for the collection of SLD. The larger sample area of the 12 mm cone also represents an improvement over the 8 mm cone, which is especially relevant when measuring FZDZ, FZRA or mixed-phase conditions in natural clouds, where very few large particles are present and a short sampling time is crucial.

*Data availability.* The complete data sets from the BIWT and from RTA are available under: https://doi.org/10.5281/zenodo.6817112 (Lucke et al., 2022). Collins IWT considers raw measurements from its tunnel as confidential information, therefore the data cannot be made public.

*Author contributions.* JL prepared the manuscript, operated the DLR probes at Collins and in the BIWT and was responsible for the overall data analysis. TJW assisted in developing the concept for the manuscript, the discussion of the data evaluation and the planning and execution of the IWT campaigns. MH, WB and VRB conducted the measurements at Collins, RTA and the BIWT respectively and provided the wind tunnel data. RH and VH operated the DLR probes during the measurements at RTA. MM assisted with the analysis of the CIP data. TJW and CV designed the experiment. All authors commented on the manuscript.

*Competing interests.* The authors declare no competing interests.

*Acknowledgements.* We thank Galdemir Botura and the technical staff of Collins Aerospace for supporting icing innovation and for the constructive and inspiring collaboration within the SENS4ICE project. We thank Embraer for providing their Cloud Combination Probe and Matt Freer for his support in the measurement campaign at the Collins IWT. We also appreciate the help and expertise of Stephan Sattler during the wind tunnel measurements at the BIWT and the support of Inken Knop in the data evaluation. This project has received funding from the European Union's Horizon 2020 research and innovation programme under grant agreement n° 824253 (SENS4ICE). The research in the RTA climatic wind tunnel was conducted as part of the ICE GENESIS project, which has received funding from the European Union's Horizon 2020 research and innovation programme under grant agreement n° 824310.



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
