# Peer review of "Icing Wind Tunnel Measurements of Supercooled Large Droplets Using the 12 mm Total Water Content Cone of the Nevzorov Probe"

_EGUsphere, 2022_

## Referee Comment (RC2)

**Review of "Icing Wind Tunnel Measurements of Supercooled Large Droplets Using the 12 mm Total Water Content Cone of the Nevzorov Probe" by Lucke et al.**

**Overview**

This study is focused on characterizing the performance of the 8mm and 12mm Nevzorov TWC cone sensors in Appendix C and Appendix O conditions. The measurements were performed across three icing wind tunnels, thus enhancing the value of this study. The results obtained in this work are of great importance for the characterization of SLD sprays, which remain one of the challenges in ground based icing test facilities. I did not find any major issues in this study. The paper is well written and it should be accepted for publication after addressing a few minor comments listed below.

**Recommendation**: The paper should be published in AMT after minor revisions.

**Comments**

1. One of the problems of microphysical measurements in icing wind tunnels is the spatial non-uniformity of sprays across the test section. This may result in biases of MVD and/or LWC measurements conducted by different instruments if their sampling volumes are positioned at different locations. To mitigate this problem, researchers usually attempt to mount instruments in the same location when conducting comparisons of different instruments or calibrations. The authors briefly mentioned this problem. However, it is not clear what was the of the spatial inhomogeneity of the wind tunnel sprays and what was its effect on the biases of the Nevzorov measurements. Did authors attempted to estimate LWC biases between the LWC, $TWC_8$ and $TWC_{12}$ Nevzorov sensors due to the sensors spatial separation, by moving the Nevzorov sensor up and down (right and left)? Do you have any estimates of spatial inhomogeneity for each wind tunnel? Such discussion would be beneficial for the paper.

2. It would be relevant indicating that the sensor head employed in this study was designed by the Environment and Climate Change Canada (ECCC) and manufactured by SkyPhysTech Inc. This sensor was tested by ECCC in the NRC AIWT wind tunnel and then used during the In-Cloud ICing and Large-drop Experiment (ICICLE) flight operation for characterisation of icing cloud environment.

3. It appears that the authors refer to the LWC sensor as "Hotwire" throughout the text. In fact, "hotwires" are a class of sensors/instruments used for measurements of condensed water content. However, the term "hotwire" is equally applicable to the 8mm and 12mm cone TWC sensors as well. For that reason, statements, like "…*for the Hotwire and the 8 mm cone*…" sound confusing. It would be reasonable to use conventional names of the hotwire sensors, i.e. "LWC sensor" when applied to a cylindrical hot-wire sensor, and "TWC 8mm (or 12mm) cone" when talking about the TWC 8mm (or 12mm) hotwire cone sensors.

4. Line 178: Korolev at al. (1998a) was focused on studies of the formation of diffraction images of spherical particles in OAPS. However, it did not discuss size corrections of out-of-focus droplet images. This problem was studied in Korolev (2007). Therefore, Korolev at al. (1998a) should be replaced by Korolev (2007). (Korolev, A. 2007: Reconstruction of the Sizes of

Spherical Particles from Their Shadow Images. Part I: Theoretical Considerations. *Journal of Atmospheric and Oceanic Technology,* **24**, 376–389. https://doi.org/10.1175/JTECH1980.1 )

5.  Page 6: It is worth mentioning that the average value L*=2580 J g-1 in Korolev et al. 1998a was obtained for a different set of ranges of temperatures and pressures as compared to this study.

6.  It is worth providing a brief geometrical description of the Nevzorov TWC 8mm and 12mm cones, i.e. inverted cones with the apex angle 60deg and the depths of the cones (~7mm and ~10.4mm).

Alexei Korolev

---

## Author Comment (AC1)

**Response to Referee #1**

Thank you very much for your helpful and constructive feedback. We reworked the manuscript according to your suggestions. Please find below our replies:

**General comments:**

"Please clarify whether the new Nevzorov probe featuring a 12 mm cone was developed within your study presented in the manuscript. If so, I consider that as an advantage of your work. Please provide details about the geometry and design of the probe then. If not, please provide information about the producer and whether there were any tests or experiments involving this probe preceding yours."

The sensor head with the 12 mm cone was not developed as part of this study. Instead, it was developed by Environment and Climate Change Canada and manufactured by the Canadian company SkyPhysTech Inc. Prior to this study it has been tested in the Altitude Icing Wind Tunnel of the National Research Council of Canada and was flown during the In-cloud Icing and Large-drop Experiment (ICICLE). As far as we are aware, there have been no publications so far that detail the performance of the Nevzorov probe during these tests. We added the information about the manufacturer and the testing to the manuscript. Furthermore, we added an image which depicts the geometry of the new sensor head.

"Section 2 introduces two research projects and three wind tunnel facilities. In my opinion, the information about the facilities is very important in this manuscript. Therefore, the section should focus primarily on the facilities participating in the current study rather than on the scope of broader projects. I suggest emphasizing the capabilities of the wind tunnels which justify their choice in the light of the objectives of this study and the differences between the wind tunnels which explain the advantage of using three facilities instead of one."

We added information on how we selected the participating wind tunnel facilities. More information on the facilities itself can be found in the corresponding publications, e.g. Ferschitz et al. (2017), Breitfuss et al. (2019), Lang et al. (2021) for RTA, Herman et al. (2006) for Collins and Bansemer et al. (2018) and Knop et al. (2021) for the BIWT. Due to confidentiality reasons, we are unfortunately not able to publish additional information on the spray systems of the individual tunnels.

"Please explain the choice of conditions for your test cases. As stated in section 4, many of them lie outside the range specified in Appendix C and Appendix 0 and some even feature above-zero temperatures. Is such a choice motivated by the limitations of the droplet generation systems, the limitations due to sampling statistics, the intention to explore the region outside Appendix 0 to allow for a possible extension of icing safety standards in the future?"

The IWT campaigns were part of EU-projects that have the goal to test and characterize newly developed sensors (SENS4ICE) and to develop tools for the 3D simulation of SLD and snow icing conditions (ICE GENESIS). DLR was tasked to validate the wind tunnel conditions through measurements with the Nevzorov (and in some tunnels also with the CCP). Generally, the wind tunnel owners aim at creating realistic icing conditions. Most of the SDS test points were therefore intended to cover the Appendix C icing envelopes. A few of the test SDS test points at larger MVD (e.g. Collins test points C19-C30) were run specifically to characterize the collision efficiency of the Nevzorov probe. The choice of FZDZ and FZRA conditions was mostly dictated by the droplet spray systems of the wind tunnels. The generation of FZDZ and FZRA was developed or improved as part of the Sens4Ice and

ICEGENESIS projects. The ultimate goal of the wind tunnel owners is to produce icing conditions that fall into the region of Appendix O. However, the large amount of water contained in SLD makes it very difficult to adhere to the LWC requirements of Appendix O while preserving uniformity (in space and time) of the droplet spray. The examined test points represent the conditions which were regarded as the most suitable under consideration of the low LWC – spray uniformity tradeoff by the wind tunnel owners. We added this information to the manuscript. Above zero temperatures were used in a few cases to avoid ice accretion and thereby make the most economic use of the wind tunnel time, as the droplet spray is assumed to be independent of temperature. Nonetheless, even above zero temperatures can be useful for icing applications, for example if the freezing of raindrops on a structure that is colder than zero degrees is investigated.

"MVD is not a particularly meaningful measure in the case of bimodal DSD, as you pointed out in section 7. Therefore, I suggest additionally including (e.g. in Table 3, at least for bimodal cases) the parameter you actually used to distinguish between FZDZ and FZRA, i.e. the diameter corresponding to the position of the maximum of the largest mode in LWC distribution. For unimodal cases, it is presumably close to MVD because the only mode is obviously the largest one. However, for bimodal cases it might give useful information concerning SLD in the DSD."

**We added the diameter that corresponds to the position of the maximum of the large droplet mode to Table 3 for the bimodal cases.**

"Figures 5, 6, 8. The range of the vertical axis is inappropriate for the presented results. Please refine the range accordingly so that the differences in position between the datapoints are visible."

We adjusted the range of the vertical axis for the figures mentioned.

**Specific Comments:**

"Line 11. The sentence implies that the form of the curve was experimentally derived. I suggest mentioning that a specific function was assumed."

We added the information that a specific function was used.

"Line 113. Did you actually use Eq. (3) instead of Korolev's value? Please specify."

The value from Eq. 2 and 3 was used, we added an explanatory sentence to the paragraph.

"Line 124. Which particular collection efficiencies from the literature did you use for the 8 mm cone in your analysis? Please specify and provide relevant references."

We used the curves from Strapp et al. (2003) to correct the 8 mm TWC cone measurements. We applied the correction from the curve which matched the actual tunnel velocity best, i.e. for the measurements at 40, 60 and 67 ms-1 we used the curve for 67 ms-1, for the measurements at 85 ms-1 we used the curve for 100 ms-1. We added this information to the manuscript.

"Section 4.1 and 4.2. At RTA, there were multiple reference instruments applied to measure LWC and DSD. How are those measurements combined to produce final estimates? With a similar procedure as you described for CDP and CIP? And what sampling statistics is considered sufficient while selecting the threshold size (line 180)? Did you follow any method described in the literature?"

RTA produced its size distribution from the mean of the cumulative distribution that was calculated from the Malvern probe data and the cumulative distribution that was computed from the FCDP-2DS-PIP combination. For the CDP-CIP combination we required that at least one particle of a size bin is measured every five seconds, otherwise we considered the particle count in this bin to be too small and switched to the CIP size distribution. Assuming Poisson statistics and the minimum time of 1.5 minutes that was used to record a size distribution, this yields a maximum uncertainty of 24% in that respective bin. In most cases, size distributions were recorded over at least four minutes, which reduces the maximum uncertainty in the last bin of the CDP due to sampling statistics to 14%.

"Section 4.3. As far as I understand, the estimated uncertainties of LWC measurements are valid only in the size range corresponding to Appendix C conditions, i.e. small droplets. Did you find any quantitative information about the uncertainties in SLD conditions?"

Unfortunately, we are not able to quantify the uncertainties in SLD conditions. Referring to an extended discussion of the uncertainties of the Nevzorov probe, that we now included in the Appendix, we note, that neither the intrinsic uncertainties of Nevzorov probe nor the uncertainties in the convective heat loss term depend on the type of droplet spray that is produced. Errors related to the collision efficiency of droplets will even be smaller in SLD conditions than in SDS conditions, because the collision efficiency of SLD is essentially 100%. In our opinion, there are two factors that increase the uncertainties in SLD conditions: First, the effect of droplet splashing is unknown. The Nevzorov sensors were designed to mitigate splashing effects. On the basis of high-speed camera images, Korolev et al. (2013) claim that the amount of ice particles which bounce from the 8 mm cone is small. The design of the new 12 mm cone is even better than that of the 8 mm cone for retaining ice particles and droplets. All this suggests, that the influence of droplet splashing effects is rather small, but at this point we cannot quantify the exact magnitude. The second source of uncertainty is caused by high frequency flutter of the sensor head around its axis of rotation, which was observed to be significantly stronger during SLD conditions with high LWC than in SDS conditions. This flutter led to (very short term) deflections of the sensor head of up to  $\pm 20^{\circ}$ . The change in sample area caused by the flutter is however just a few percent.

"Line 191. I assume here you give an estimate of the uncertainty of LWC measurements with a 8 mm collector cone. Please clarify."

The ±15% provided in the text were a rough estimate for both the LWC sensor and the 8 mm cone that was estimated on the basis of the convective heat losses that we observed and the accuracy estimates from the manufacturer of the probe. In response to your comment we performed a more thorough estimation of the measurement uncertainties, which can now be found in the Appendix.

"Section 4.4. The number of test points documents extensive experimental work. Please consider whether it would be helpful for the reader to conceive the range of conditions explored if the test points and the regimes (SDS, FZDZ, FZRA) are illustrated in a figure, e.g. a scatter plot LWC vs. diameter of the largest mode (the parameter mentioned above in general comment #4 which you used to distinguish between FZDZ and FZRA). The overall LWC limits of Appendix C and Appendix 0 can then be marked for the respective regimes."

**We thank the reviewer for this suggestion, we added a plot which depicts the different test points as a function of LWC, MVD and the icing regime.**

"Lines 258-259 and Fig. 4. What collision efficiency correction did you use? Please provide a reference. Is such a selection of the sensor depending on MVD recommended in existing literature? If so, please cite a relevant source."

We used the collision efficiencies from Langmuir et al. (1946) for the LWC sensor and from Strapp et al. (2003) for the 8 mm TWC cone. An explanation regarding the applied collision efficiencies has been added to section 3. The justification for using the LWC sensor up to an MVD of 20  $\mu$ m is derived from

Schwarzenboeck et al. (2009) which states: "A maximum in  $\varepsilon_{LWC,droplets}$  is reached roughly around 20– 30  $\mu$ m, indicating that droplets smaller than 20–30  $\mu$ m partly tend to curve around the LWC sensor, whereas larger ones impact with decreasing efficiencies related to a loss in droplet mass.  $\varepsilon_{LWC,droplets}$ rapidly starts to decrease (with increasing droplet size) beginning at droplet sizes beyond 30–40  $\mu$ m."

"Table 5. Providing a 2 sigma interval is rather unusual. Typically, just 1 sigma is reported and it is understood in the context of estimated standard deviation of the distribution of the results. This remark regards reporting of uncertainties and does not interfere with the point you make in line 285 where even the 3 sigma test criterion can be applied."

We now report the 1 sigma interval for the parameter  $D_0$ .

"Line 286. Please specify explicitly how you calculate LWC\_12. Is it just measured LWC multiplied by a factor f(MVD) or does the computation involve DSD spectrum?"

The collision efficiency used for  $LWC_{12}$  was computed using the full DSD. We added a sentence which clarifies this.

"Line 304. How do you know that droplet coincidence was present? Is it simply implied by the high droplet concentration?"

Droplet coincidence was detected through an analysis of particle transit times. We added a section in the Appendix which describes the analysis that we performed.

"Section 7. I suggest modifying the section title to mark contrast to section 6, e.g. (Application of collision efficiencies in bimodal SLD conditions) or similarly."

We changed the section title as you suggested, thank you for the nice proposal.

"Lines 350-354 and Figure 8. Please specify explicitly whether those results were obtained by applying the MVD approximation or by resolving the entire DSD."

Again, the full DSD was used to compute LWC12 and LWC8. A sentence clarifying this was added to the paragraph above.

"Table 6. The values of epsilon\_12 are not explained and commented on in the text. Do they result from the integration of DSDs multiplied by collision efficiency curve or represent a value f(MVD) of MVD approximation? If the latter, please explain why they do not agree with Fig. 4."

The values of  $\varepsilon_{12}$  are derived using the full DSD. We modified Eq. 9, so that this is apparent now.

**Minor issues:**

We corrected all the minor issues that you pointed out. Below are some comments on specific issues that needed clarification:

"Lines 34 and 36. I assume "they" refers to the last citation given in the text. Then another citation at the end of the sentence is confusing. Please be specific about which reference you mean."

Lines 34 and 36: The publication "Characterization of Aircraft Icing Environments with Supercooled Large Drops for Application to Commercial Aircraft Certification" by Cober and Isaac (2012) is largely based on the FAA report "Data and Analysis for the Development of an Engineering Standard for Supercooled Large Drop Conditions" by Cober et al. (2009) and the statements in lines 30-37 can be derived from either publication. We agree with the reviewer that the citations in lines 35 and 37 were

confusing and hence removed them, now it should be clear that the findings originated from the 2009 FAA report by Cober et al.

"Table 3. The last FZDZ record for BIWT. Should the temperature be +5 deg or a star is erroneously given here?"

The temperature was -5°C, the star was erroneously given and has been removed.

"Line 250. According to Table 4, Group 1 contains measurements from two wind tunnels. Please clarify."

Group 1 contains measurements from two wind tunnels as stated in Table 4. The sentence in line 250 has been corrected.

---

## Author Comment (AC2)

**Response to Alexei Korolev (Referee #2)**

Thank you very much for your constructive feedback and your suggestions for improvements. We revised the manuscript and adapted the changes that you proposed. You find our reply to each of your comments below:

1. "One of the problems of microphysical measurements in icing wind tunnels is the spatial nonuniformity of sprays across the test section. This may result in biases of MVD and/or LWC measurements conducted by different instruments if their sampling volumes are positioned at different locations. To mitigate this problem, researchers usually attempt to mount instruments in the same location when conducting comparisons of different instruments or calibrations. The authors briefly mentioned this problem. However, it is not clear what was the of the spatial inhomogeneity of the wind tunnel sprays and what was its effect on the biases of the Nevzorov measurements. Did authors attempted to estimate LWC biases between the LWC, TWC8 and TWC12 Nevzorov sensors due to the sensors spatial separation, by moving the Nevzorov sensor up and down (right and left)? Do you have any estimates of spatial inhomogeneity for each wind tunnel? Such discussion would be beneficial for the paper."

We generally attempted to measure the droplet spray at the same position with all sensors. However, as you mention, this is not possible for all the Nevzorov sensors, due to their spatial separation. At the BIWT, we therefore performed traverse measurements to find an area with a homogeneous spray distribution. The area that we determined (which extended from the lowermost Nevzorov sensor to the uppermost Nevzorov sensor) had an LWC homogeneity of ±3% in bimodal conditions. Collins also provided information on its tunnel inhomogeneity which shows that within the area spanned by the Nevzorov sensors, both the small droplet spray and the FZDZ spray are uniform within ±10%. For RTA, uniformity measurements presented in Breitfuss et al. (2019) and further internal tunnel calibrations show that LWC deviations in the center of the tunnel cross section where the Nevzorov sensors were positioned are no larger than ±5% for both FZDZ and FZRA conditions.

2. "It would be relevant indicating that the sensor head employed in this study was designed by the Environment and Climate Change Canada (ECCC) and manufactured by SkyPhysTech Inc. This sensor was tested by ECCC in the NRC AIWT wind tunnel and then used during the InCloud ICing and Large drop Experiment (ICICLE) flight operation for characterisation of icing cloud environment."

We agree that information on the manufacturer and testing prior to our study is important and we added the provided information to the manuscript.

3. "It appears that the authors refer to the LWC sensor as "Hotwire" throughout the text. In fact, "hotwires" are a class of sensors/instruments used for measurements of condensed water content. However, the term "hotwire" is equally applicable to the 8mm and 12mm cone TWC sensors as well. For that reason, statements, like "…*for the Hotwire and the 8 mm cone*…" sound confusing. It would be reasonable to use conventional names of the hotwire sensors, i.e. "LWC sensor" when applied to a cylindrical hot-wire sensor, and "TWC 8mm (or 12mm) cone" when talking about the TWC 8mm (or 12mm) hotwire cone sensors."

We agree that the name "Hotwire" is misleading and replaced the term by "LWC sensor" throughout the text.

4. "Line 178: Korolev at al. (1998a) was focused on studies of the formation of diffraction images of spherical particles in OAPS. However, it did not discuss size corrections of out-of-focus droplet images. This problem was studied in Korolev (2007). Therefore, Korolev at al. (1998a) should be replaced by Korolev (2007). (Korolev, A. 2007: Reconstruction of the Sizes of Spherical Particles from Their Shadow Images. Part I: Theoretical Considerations. *Journal of Atmospheric and Oceanic Technology,* **24**, 376–389. https://doi.org/10.1175/JTECH1980.1 )"

Thank you for pointing this out, we changed the reference according to your suggestion.

5. "Page 6: It is worth mentioning that the average value L*=2580 J g-1 in Korolev et al. 1998a was obtained for a different set of ranges of temperatures and pressures as compared to this study."

We added a sentence that explains that the value of 2580 J/g was derived for aircraft measurements where different temperatures and pressures prevail.

6. "It is worth providing a brief geometrical description of the Nevzorov TWC 8mm and 12mm cones, i.e. inverted cones with the apex angle 60deg and the depths of the cones (~7mm and ~10.4mm)."

We added a figure which details the dimensions of the new Nevzorov sensor head.